# Analysis and Assessment of the Usefulness of Recycled Plastic Materials for the Production of Airfield Geocell

**DOI:** 10.3390/ma14133557

**Published:** 2021-06-25

**Authors:** Mariusz Wesołowski, Piotr Włodarski, Paweł Iwanowski, Agata Kowalewska

**Affiliations:** Air Force Institute of Technology, Airfield Division, ul. Ks. Bolesława 6, 01-494 Warsaw, Poland; piotr.wlodarski@itwl.pl (P.W.); pawel.iwanowski@itwl.pl (P.I.); agata.kowalewska@itwl.pl (A.K.)

**Keywords:** plastics, recycling, natural airfield pavement, safety of air operations

## Abstract

The load-bearing capacity of natural airfield pavement has a direct impact on the safety of air operations. Unfortunately, the field tests often indicate that the load-bearing capacity of natural airfield pavements is not sufficient. In this case, it is necessary to reinforce them in order to meet the requirements set out in international documents. It is important that the method of reinforcing the subsoil is fast and as noninvasively as possible. There are many methods of reinforcing the subsoil, however, they are expensive and time-demanding, which involves turning off the airport for a long time. Airfield geocells made of recycled plastics discussed in the article seem to be the optimal solution due to the quick implementation of their application by pressing into the existing natural pavement. The article presents the results of laboratory tests demonstrating that material in question is load-resistant and chemical-resistant, while field studies have confirmed that the airfield geocell made of the plastic in question improves the load carrying capacity of natural airfield pavement.

## 1. Introduction

### 1.1. Overview

All over the world, aviation infrastructure is extensive on a huge scale, including civil, military, aeroclub, exclusive use, airstrikes and helicopters. Each of these facilities must meet specific requirements regarding the condition of the airfield pavement, which has a direct impact on the safety of air operations [1].

Airfield pavement is any type of adequately prepared pavement on which aircraft can move [2]. Depending on the needs and type of airport functional element, the following types of airfield pavements are distinguished:Rigid pavements;Flexible pavements;Semi-rigid pavements;Natural pavements.

This article dealt with the issue of natural airfield pavement. Natural airfield pavement occurs as ground and turf. They are prepared by a properly made ground substrate, which ensures the safe movement of the aircraft without damaging its design. On aerodrome facilities with artificial pavement intended for air operations, natural pavement may occur on the following airport functional elements:Runway shoulders;Runway end safety areas (RESA).

Runway shoulders shall be prepared or constructed in such a way that, in the event of the aircraft being run off the runway, they can carry the weight of the aircraft without damaging the aircraft structure and that they can carry the weight of ground vehicles which may move on the side of the road [3]. The runway end safety areas should be prepared or constructed in such a way as to minimise the risk of damage to aircraft that has touched down in front of the runway or off the runway, to increase the possibility of braking the aircraft and to allow the movement of rescue and fire-fighting vehicles [3]. The most important parameter characterizing the natural airfield pavement is its load carrying capacity, which is determined by the California Bearing Ratio CBR. According to [4] the California Bearing Ratio is the percentage of force needed to push the standardized piston into the test ground to a certain depth compared to the force needed to push the piston to the same depth in the standardized soil sample. The load capacity of the natural aerodromes is, according to [5], the ability of the pavement to carry a specific load from an aircraft without risk of damage.

Previous studies by the authors of this article show that the load bearing capacity of natural airfield pavement often does not meet certain requirements, which can be a direct result of a plane crash. This was the case on 17 July 2007, when an Airbus A320–233 veered off the runway at Congonhas Airport (Brazil) and hit a petrol station and a building. In this case, 187 people were killed, including the crew. In another case recorded on 22 May 2010 a Boeing 737–800 plane crash landed in severe weather conditions. The plane did not stop on the runway, slid into the valley and went up in flames–158 people died [6].

In order to improve the load capacity of natural airfield pavements, different methods of reinforcing the ground substrate are used [7].

The article will describe a method of reinforcing natural airfield pavements with an airfield geocell made of recycled plastic materials.

### 1.2. Overview of Existing Solutions

Airfield geocells can be classified as geosynthetics in a very general way; they are largely used to strengthen the subsoil. Geosynthetic, according to [8], is a general term for which at least one component is made of synthetic or natural polymer, in the form of a sheet, tape or spatial form, used in contact with land and/or other materials in geotechnicals and construction. An example may be the case described in the article [9], which involved the use of a two-way reinforced composite foundation made of a mattress reinforced with geocells and gravel piles. According to the American Association for Research and Materials [10], geosynthetics are flat products made of polymeric material, used with soil, rocks or other materials related to geotechnical engineering as an integral part of a design, structure or system. However, the element (airfield geocell) that is the subject of this article does not appear in the geosynthetics qualification according to [8], but due to its function and composition it can be classified as a group of geosynthetics.

According to the PN-EN ISO 10318: 2007 [8] standard, geosynthetics are divided into geotextiles, geotextile related products, geosynthetic barriers and geocomposites. Each type of geosynthetics has its own application, including reinforcement weak subsoil, securing slopes against erosion, and as separation and drainage layers. These geosynthetics are part of the pavement structure. It is connected with the necessity to reconstruct the existing pavement and use a geosynthetic in accordance with the design of the structure of the layers of the structure. This entails the need to shut down the airport for a longer period of time. The main advantage of using airfield geocells made of plastic is the ability to quickly restore the operational capacity of the airport because they are rolled directly into the existing natural pavement.

The authors of [11], which describe the possibilities of reinforcement of ground routes in wet conditions, geocells (such as in this work) have been classified as Geo-Others–Turf Reinforcement (Figure 1). Geo-Others are described in [11] as reinforcements made of recycled plastics designed to protect the turf from rut formation, against soil erosion and to support turf compaction. In [12], the author describes geosynthetics as part of the group of geo-others that geocells also qualify for [13].

In [14], the authors presented the results of studies on the effect of the use of reinforcement of verges of unpaved cellular geosynthetics on vegetation growth. During the test period, it was not noticed that the geocells limited the growth of vegetation. Various combinations of substructure and the topsoil layer were studied, which turned out to have a significant impact on vegetation growth during observation.

The main group of raw materials from which the airfield geocell discussed in the article is produced comes from recycled plastic material. 

In national and foreign literature, you can read many studies that describe laboratory experiments involving the use of cellular geogrid in load-bearing structural layers. The article [15] describes a series of static and cyclic plate load tests performed by a research team at the University of Kansas on geocell-reinforced primers with various filling materials, i.e., poorly sorted sand from the Kansas River, quarry waste and recycled asphalt pavement. Studies have shown that the geocell used to strengthen the substructures increased the load carrying capacity and rigidity of the substrates, reduced permanent deformation, reduced vertical stresses. The use of cellular geogrid has also affected the possibility of reducing the required thickness of the substructure to achieve the same parameters as the road on weak ground [15].

Geocells are used successfully as reinforcement for road and airfield pavement structural layers. The article [16] presents results that confirmed that the use of cellular restraining systems in the road substructure layer allowed for a reduction in the thickness of structural layers by up to 50% compared to the road without the use of geocell armament. The use of plastic geocells as reinforcement of structural layers or existing natural pavement makes it possible to use local filling materials, which generates economic (reduced investment costs) and environmental conditions due to reduced earthworks, which in turn reduces fuel consumption, pollution from vehicles [16].

Companies involved in the production of road geocells, which are used on access roads or parking lots, are increasingly developing new geogrid technologies that they use to strengthen natural airfield pavements. Novus HM proposes to strengthen the airport’s grassy pavement by using the TERRA-GRID^®^ E 35 (Figure 2) geocell [17]. It is a product made of plastic polyethylene (PE)/polypropylene (PP), which provides a load capacity of up to 160 t/m^2^ (depending on ground conditions and soil preparation). Novus HM declares that the TERRA-GRID^®^ E 35 geocell is resistant to UV radiation, frost, oils, solvents, salt and most acids.

PERFO has developed a ground reinforcement system in the form of geocells for reinforcing various pavements, including grassy airfield pavements (Figure 3). The PERFO geocell is made of plastic polyethylene (PE)/polypropylene (PP), which provides a load capacity of about 60 tons/m^2^ (depending on ground conditions and substrate preparation). PERFO geocells have been successfully used at many airport facilities to increase the safety of air operations [18].

Narew Airport 2 in Poland has a runway with grass pavement paved with geocells made of plastic and a length of 1500 m (Figure 4). Of the runway built in this technology, the runway in Narew is the longest runway in the world built with geogrid technology [19].

### 1.3. Recycled Plastic Material as Airfield Geocell Material

Structural plastics, e.g., polyethylene, polypropylene and polycarbonate, are used in various areas of life [20], such as engineering, electronics or electrical engineering, mainly due to their excellent thermal stability and high heat deformation temperatures [21].

This airfield geocell was formed by injection method of polyethylene and polypropylene obtained from the recycling process of plastic waste.

Polyethylene (PE) is an ethylene polymer with a repeatable structural unit of the main chain [CH_2_–CH_2_] [20]. Polyethylene, depending on the conditions under which polymerization takes place, is divided into:High-pressure polyethylene–low density (e.g., LDPE);Low-pressure polyethylene–high density (e.g., HDPE).

Low-density polyethylene–high pressure (LDPE) is obtained from ethylene in the gas phase at a pressure of 180–250 MPa and at a temperature of 200–250 °C [22]. The density of high-pressure polyethylene is between 0.90 and 0.94 g/cm^3^ [23]. 

High density polyethylene (HDPE) is formed during a liquid phase polymerization reaction at 50–70 °C [22]. Chemically, it is closest to pure polyethylene (Figure 5). The density of low-pressure polyethylene is approximately 0.94–0.97 g/cm^3^ [23].

Polypropylene is formed by polymerization of propylene [24]. Polymerization of polypropylene is usually carried out in a solution at a temperature of 50 °C to 100 °C. Polypropylene is one of the lightest plastics, with a density between 0.85 and 0.92 g/cm^3^ [22].

### 1.4. Injection Method of Forming Finished Products from Recycled Plastic Waste

The growing demand for plastics requires the development of new technologies for their manufacture, processing and modification. Among the processing methods of thermoplastics, the injection process is a popular method [25]. Injection formatting is a leading technique for producing complex polymer elements. Demanding designs of plastic products, high quality requirements and time constraints force the optimisation of various input parameters crucial for achieving the desired quality indicators [26].

Plastic injection is a cyclic process of manufacturing polymer products, which involves melting the plastic, most often granules, and then being fed through the nozzle into the mould cavity. In pressurised form, the material enters a solid state and is removed as a finished product [27]. The diagram of the automated injection socket and the injection moulding machine construction diagram are shown in Figure 6 and Figure 7 [28].

## 2. Materials and Methods 

The plastic from which the airfield geocell was made has been subjected to material tests to determine physico-mechanical properties and to test its resistance to chemical agents used on airfield pavement during their year-round operation.

The finished technology of strengthening the natural airfield pavement with the finished product of an airfield geocell made from recycled plastic waste has been tested by a training ground.

### 2.1. Static Flexural of Plastic Samples

Static bending tests of plastic samples were carried out in accordance with PN-EN ISO 178:2019-06 Plastics. Determination of flexural properties [29]. The three-point bending tests were performed on eight rectangular samples (including three test samples) injected plastic measuring approximately 80 mm × 10 mm × 4 mm. Figure 8 shows a view of the sample fixed in the strength machine before the test, and Figure 9 presents the view of the sample fixed in the strength testing machine after stopping the test [30].

### 2.2. Static Tensile of Plastic Samples

Static tensile testing of plastic samples has been carried out in accordance with PN-EN ISO 527-1:2020-01 Plastics. Determination of tensile properties. Part 1: General principles [31].

Static tensile tests were carried out on five samples, injection moulded plastic, with a total length, a measuring section width and thicknesses of approximately 170 mm × 10 mm × 4 mm. An image of the sample fixed in the strength machine before the test is given in Figure 10, while the samples are placed in Figure 11 after the test has stopped [32].

The static tensile test was carried out at v = 1 mm/min.

### 2.3. Compression Test of Airfield Geocell Made of Recycled Plastic Materials

Compression studies of airfield geocell were carried out in accordance with PN-EN ISO 25619-2:2015-11 Geosynthetics. Determination of compression behaviour. Part 2: Determination of short-term compression behaviour [33].

Five airfield geocells (sample pre-test geocells—Figure 12) with dimensions of approximately 485 mm × 485 mm × 40 mm were tested and moulded plastic. Three geocells were compressed in the middle area, the other two were compressed in the corner area.

Figure 13 shows a geocell put in the strength machine before the test, while Figure 14 shows a picture of the geocell after the test [34].

The preload (zero displacement of the moving compressive plate relative to the base plate) was approximately 5 kPa corresponding to a force of 229 N.

During the test, the following parameters were determined:Compressive strength during short-term compression, σ_mr_ [kPa];Compressive strain determined from the displacement of the movable compressive plate relative to the base plate for the σ_mr_, ε_mr_ [%];Compressive strain determined by a video-extensometer for the σ_mr_, ε_mr,ve_ [%].

### 2.4. Determination of the Resistance of Plastic Samples to Consumables

Samples made of plastic were influenced by consumables, i.e., water, airfield pavement de-icing agent based on potassium formate and Jet A1 aviation fuel. The samples were completely immersed in the individual media (Figure 15, Figure 16 and Figure 17) and kept in there for a period of 14 days. The weight of the samples was measured on the individual days after they were dried with filter paper.

The effect of individual media on the test plastic was determined by calculating the absorbability of the individual samples based on a change of their weight. Absorbability *X* [%] was calculated according to the Equation (1):(1)$$X=\frac{m_{1}-m}{m}\cdot 100$$
where:

*m*—weight of the sample before immersion, [g];

*m*_1_—mass of the sample after removal from the specified medium, [g].

### 2.5. Method of Installation of Airfield Geocells on Natural Airfield Pavement

The airfield geocell was installed by pushing it into the natural airfield pavements.

The airfield geocells are arranged on a designated area on experimental field. Vehicles with a weight of 6–10 tonnes are used to push the geocells. Rolling begins perpendicular to the row of geocells. Figure 18 and Figure 19 shows the process of laying and pressing the airfield geocell into the natural pavement.

### 2.6. Tests on the Load Capacity of the Natural Airfield Pavement Reinforced with an Airfield Geocell Made of Recycled Plastic Material

The HWD (Heavy Weight Deflectometer) airfield deflectometer was used to assess the load capacity of airfield pavement; Figure 20 shows the load capacity of the natural airfield pavement reinforced with airfield geocell. The test shall measure the elastic deflections of the test pavements formed under dynamic load on a discharge basis with a force of approximately 200 kN, on a pressure plate with a diameter of 0.45 m, resting on the pavement and being carried out in accordance with defence standard NO-17-A500:2016 [35]. During the measurement, the deflections of the test pavement shall be recorded by geophones mounted on the measuring strip and centrally under the load plate of the device. The maximum distance of the measuring point from the centre of the loading plate shall be 2.5 m [36]. The results are recorded on your computer while illustrating deflection and stress waveforms and stresses over time on the monitor screen.

The training ground tests for the load capacity of natural airfield not reinforced and the load capacity of the natural airfield pavement reinforced by the airfield geocell were carried out on four experimental plots.

On the basis of the deflection bowl and knowledge of the thickness of the layers of construction and the characteristics of the materials from which they are made, the elasticity modules of the individual layers are determined [36].

The result of surface measurements using the HWD airport deflectometer is the maximum elastic deflection values measured by geophones. This set of values is defined as the deflection bowl. The size of the deflections throughout the bowl is a relationship that can be described by the (2) [37]:(2)$$U_{i}=f\left(h,\;E,\;\nu \right)$$
where:

*U_i_*—the deflection value of the test surface at the point of *I*;

*f*—function relation of the components;

*h*—thickness of the particular pavement structural layers;

*E*—elasticity modulus of the particular structural layers of the pavement and subsoil;

*ν*—Poisson’s number of the pavement and subsoil’s structural layers.

On the basis of the recorded values of the airport pavement’s deflection, the elasticity modules of the materials of the particular layers are determined by iterative comparison of the values of the measured deflections and theoretical deflections, so that function *F* has a minimum value. For this purpose, the following relationship is used (3) [37]:(3)$$F=\overset{k}{\underset{j=1}{\sum }}{\left(w_{j}-u_{j}\right)}^{2}$$
where:

*F*—approximation function of actual and theoretical values;

*w_j_*—calculated pavement deflections at a distance of r from the centre of a loading plate;

*u_j_*—measured pavement deflections at a distance of r from the centre of a loading plate;

*k*—number of geophones.

In the future, works are also planned to determine the usefulness of research on natural pavements with the use of GPR. It is aimed at a more detailed analysis of the anomalies in the ground subsoil on natural airfield pavements [38].

## 3. Results

### 3.1. Static Flexural of Plastic Samples

The results obtained for the determination of the elasticity module at flexural *E_f_* for a population of 8 samples are shown in Table 1. The flexural elasticity module has been calculated according to the Equation (4):(4)$$E_{f}=\frac{\sigma_{f2}-\sigma_{f1}}{\varepsilon_{f2}-\varepsilon_{f1}}$$
where:

*σ**_f_*_1_—is the flexural stress, measured at deflection s_1_, [MPa];

*σ**_f_*_2_—is the flexural stress, measured at deflection s_2_, [MPa].

**Table 1 materials-14-03557-t001:** Results of determination of the elasticity module at flexural E_f_ for a population of 8 samples.

Sample Number	E_f_ ^1^ [MPa]	E_fśr_ [MPa]	S_(Ef)_ ^2^ [MPa]	m_Ef_ ^3^ [MPa]	E_fMNK_ ^4^ [MPa]	E_fMNKśr_ [MPa]	S_(EfMNK)_ ^5^ [MPa]	m_EfMNK_ ^6^ [MPa]
1	946	947	14.8	935 < m_Ef_ < 959	940	944	14.1	932 < m_EfMNK_ < 956
2	935				937			
3	926				925			
4	938				938			
5	953				948			
6	973				973			
7	945				940			
8	959				952			

^1^ modulus of elasticity in flexure, flexural modulus; ^2^ standard deviation of the flexural elasticity module; ^3^ 95% two-sided confidence interval of average values of the flexural elasticity module; ^4^ flexural modulus determined from the slope of the regression line determined by the least squares method; ^5^ standard deviation of the flexion elasticity module (regression), ^6^ 95% two-sided confidence interval of average values of the flexion elasticity module (regression).

Results of flexural stress determination at conventional bend σ_fC_ for a population of 8 samples are shown in Table 2.

Graph of stress σ_f_ to strain ε_f_ (marked: points for determining the elasticity module at flexural, stress at conventional bend and maximum stress at deformation ε_f_ = 5%) is shown in the Figure 21.

During the static flexural test, the plastic samples were not broken before reaching the contractual value of the S_C_ deflection arrow (conventional deflection). In this case, the size of the test material in terms of ability to carry bending loads is the stress at a specific deflection arrow σ_fC_. This is the highest normal (flexural) stress in the sample when deflection is reached S_C_.

### 3.2. Static Tensile of Plastic Samples

The results obtained for the test samples are shown in tables: Table 3—determination of the tensile elasticity modulus E_t_; Table 4—Poisson’s number determination µ; Table 5—maximum stress σ_m_ determination; Table 6—strain at maximum stress ε_tm,ve_ determination; and Table 7—stress at the end of the test σ_b’_ determination.

A graph of stress σ on the strain ε_l_ for one of the samples is shown in Figure 22, while the stress σ relation to the strain ε_t,ve_ graph is shown in Figure 23.

The test samples were not broken, a sample rupture or elongation of the sample measured using an elongated extensometer equal to that of the 4 mm.

### 3.3. Compression Test of Airfield Geocell Made of Recycled Plastic Materials

The results obtained for the test samples are presented in tables: Table 8—results of determination of compressive strength σ_mr_; Table 9—compressive strain results ε_mr_ determined from the displacement of the moving compressive plate relative to the base plate; and Table 10—compressive strain results ε_mr,ve_ determined by video-extensometer.

A graph of the stress σ_mr_ relation to the deformation of ε_mr_ and ε_mr,ve_ for one of the samples is shown in Figure 24.

### 3.4. Determination of the Resistance of Plastic Samples to Consumables

The results of the tests carried out are shown in Table 11, in which s is the standard deviation and v is the coefficient of variation in percentage.

The tests concluded that the test plastic is resistant to water, de-icing and aviation fuel—no signs of mass degradation were observed in the samples tested. The plastic in question shows the highest water absorption after immersion in aviation fuel, i.e., about 3%.

### 3.5. Tests on the Load Capacity of the Natural Airfield Pavement Reinforced with an Airfield Geocell Made of Recycled Plastic Material

Tests on the load capacity of the subsoil and the structural system of the subsoil and the airfield geocell pressed into it made of recycled plastic material were carried out on four experimental field.

The load bearing capacity obtained from the HWD tests are shown in Figure 25.

The graph shows the results of the modulus of elasticity for unreinforced pavement, which range from E = 110 MPa to E = 128 MPa [39]. Reinforced natural pavement with airfield geocell have a modulus of elasticity between E = 139 MPa to E = 145 MPa [39].

The load-bearing capacity of natural airfield pavement reinforced with a pushing plastic airfield geocell has improved by an average of about 20%.

## 4. Discussion

On the basis of laboratory and field tests carried out during the study, it is concluded that airfield geocell made of recycled plastic material improves the load-bearing capacity parameter of natural airfield pavement and can be used as a reinforcement.

Strength laboratory tests have shown that the recycled plastic material from which the airfield geocell is made has good mechanical properties (tensile, flexural, compression) and is resistant to consumables used at airports, i.e., aviation fuel, de-icing agents and water. The greatest water absorbability of the material is after immersion in aviation fuel, i.e., about 3%.

During the strength tests of the plastic, the tensile elasticity modulus was 913 MPa and the maximum stress was 19.8 MPa. The flexion modulus was 947 MPa, while the maximum flexural stress at 5% deformation reached 20.6 MPa. The compressive strength at the central point of the airfield geocell reached 4811 kPa. 

This airfield geocell was made by injection method of polyethylene chemical called 1-butene, polymer with ethene. Typical plastic properties are shown in Table 12. The manufacturer of the plastic in question declares that it is resistant to acids, lyes and alcohols.

The load-bearing capacity of natural airfield pavement reinforced with a pushing plastic airfield geocell has improved by an average of about 20%.

In conclusion, recycled plastic material is useful for the production of airfield geocell. Due to its strength characteristics, the airfield geocell shall not be damaged or destroyed after application of the load [48]. Plastic is highly resistant to consumables such as water, aviation fuel and de-icing agents, which is very important as they are used on airfield pavements.

The main advantage of using plastic airfield geocells is the ability to quickly restore the operational capacity of the airport. The use of recycled materials is a more advantageous solution in financial terms compared to the construction of, e.g., rigid concrete pavement. Consequently, although it is a plastic it has a lower negative impact on the environment in comparison to the cement concrete production.

## 5. Conclusions

Research on the use of an airfield geocell to strengthen natural airfield pavements was started in 2018 at one of the aeroclub airports in Poland. Research on the load-bearing capacity of natural airfield pavements reinforced with an airfield geocell carried out in 2018–2020 confirmed the effectiveness of the proposed solution. The airfield geocell did not degrade after 3 years of use.

The direction of further work will include the continuation of field verification tests on real airport facilities and the improvement of load capacity and safety for military and civil aircraft with a weight exceeding 5700 kg, so that the pavement reinforced with the airfield geocell meets the load capacity requirements.

## Figures and Tables

**Figure 1 materials-14-03557-f001:**
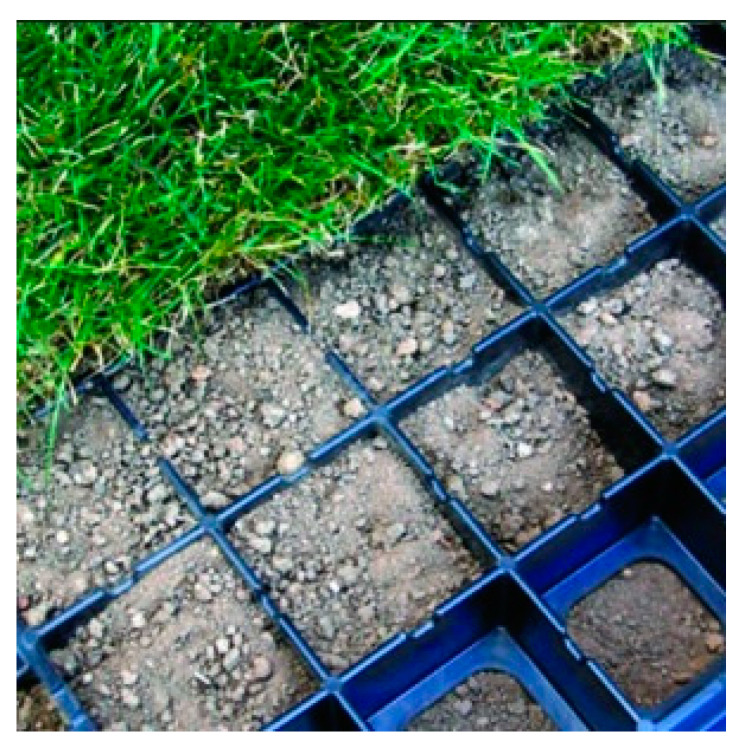
GEOBLOCK^®^ Grass Pavers.

**Figure 2 materials-14-03557-f002:**
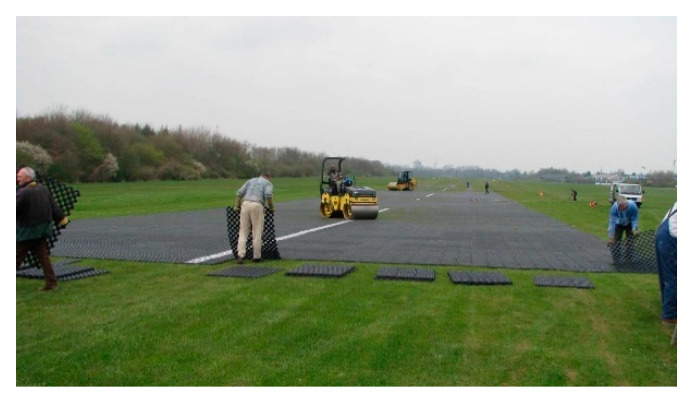
Solution made by the company Novus HM.

**Figure 3 materials-14-03557-f003:**
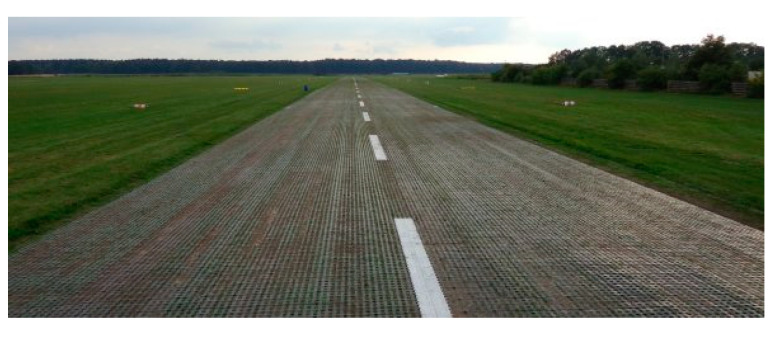
Solution made by the company PERFO.

**Figure 4 materials-14-03557-f004:**
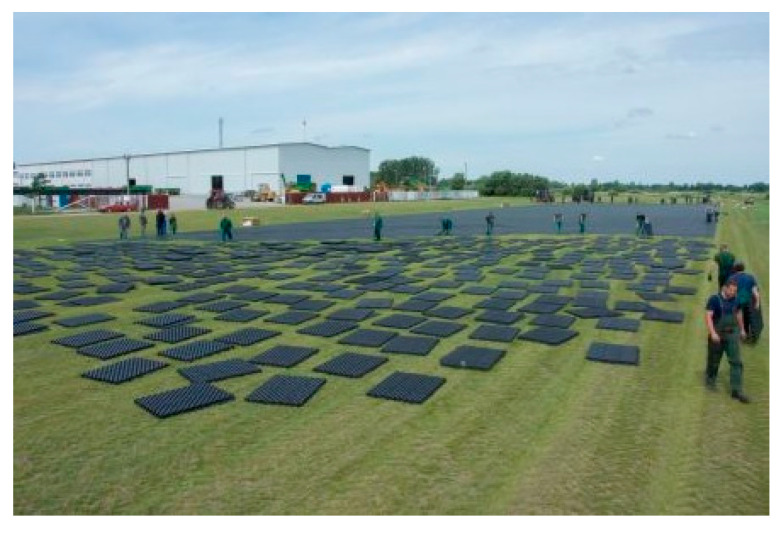
Laying the geocell on the natural pavement of the runway.

**Figure 5 materials-14-03557-f005:**
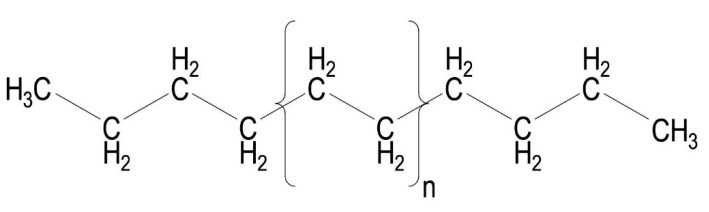
Structural formula of pure polyethylene.

**Figure 6 materials-14-03557-f006:**
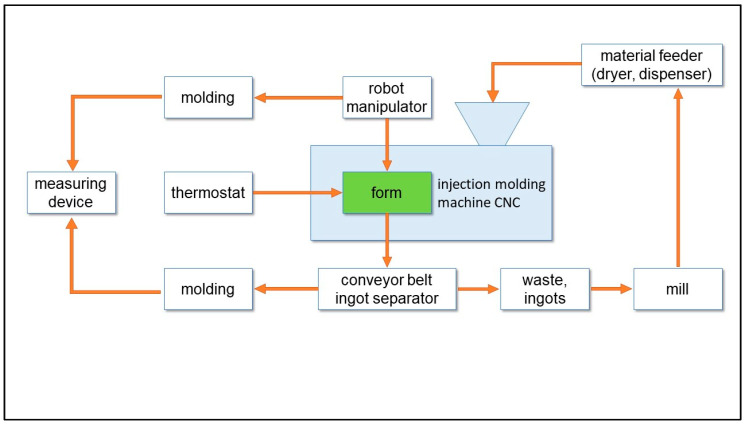
Automated production cell of injection moulding machine.

**Figure 7 materials-14-03557-f007:**
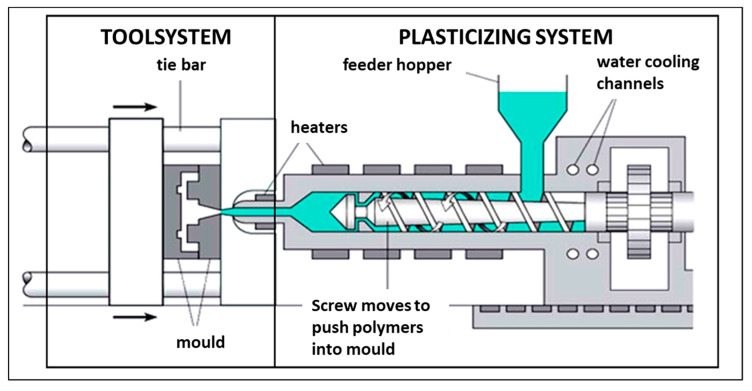
Injection moulding machine construction diagram.

**Figure 8 materials-14-03557-f008:**
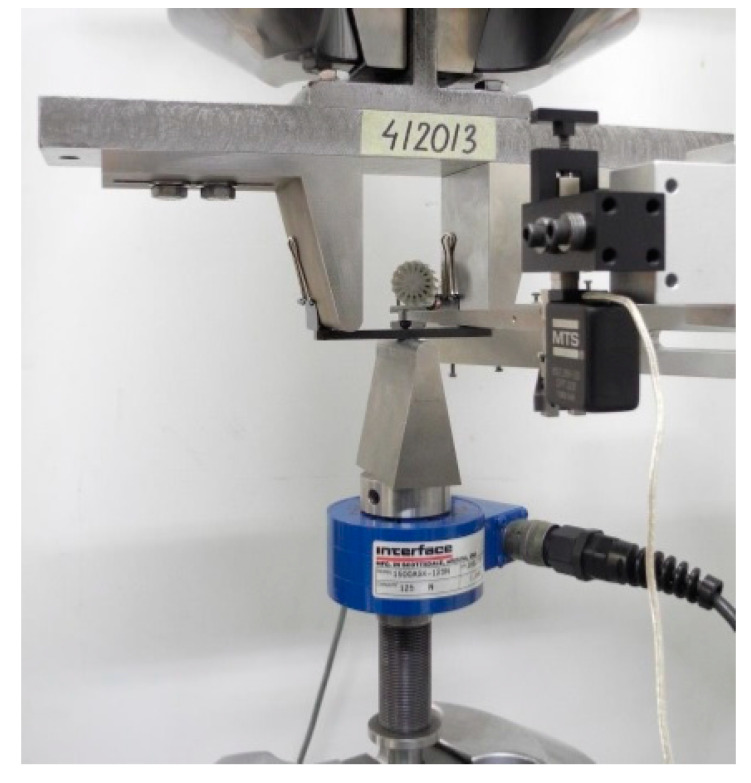
View of the sample fixed in the strength machine before the test.

**Figure 9 materials-14-03557-f009:**
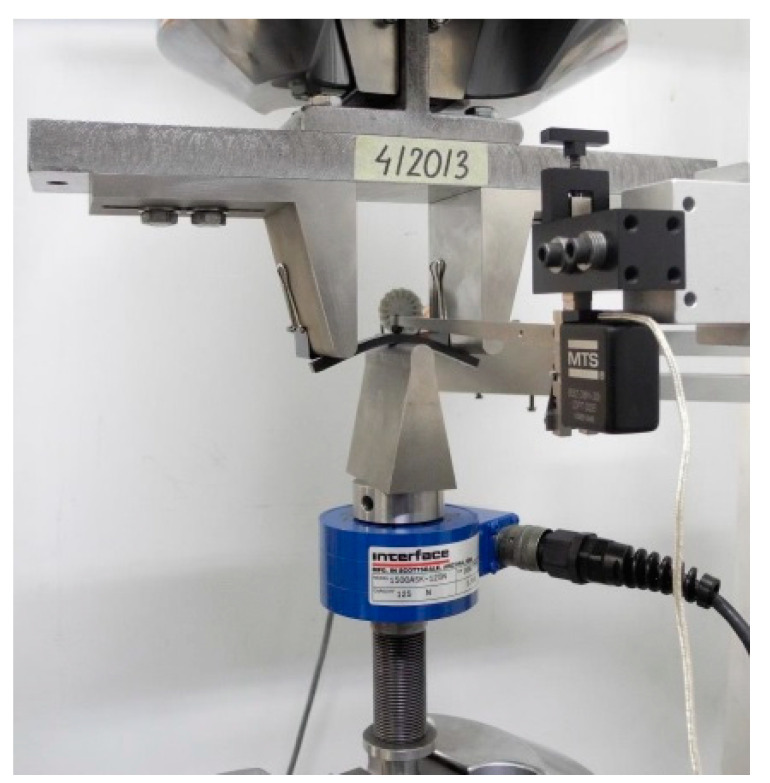
View of the sample fixed in the strength machine after stopping the test.

**Figure 10 materials-14-03557-f010:**
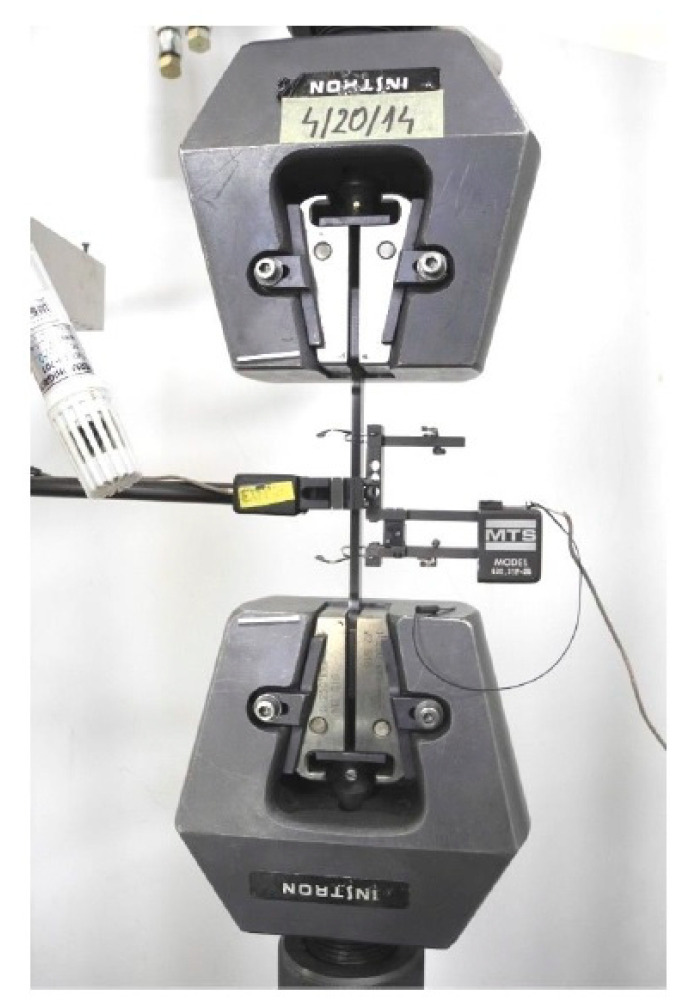
View of the sample fixed in the strength machine before the test.

**Figure 11 materials-14-03557-f011:**
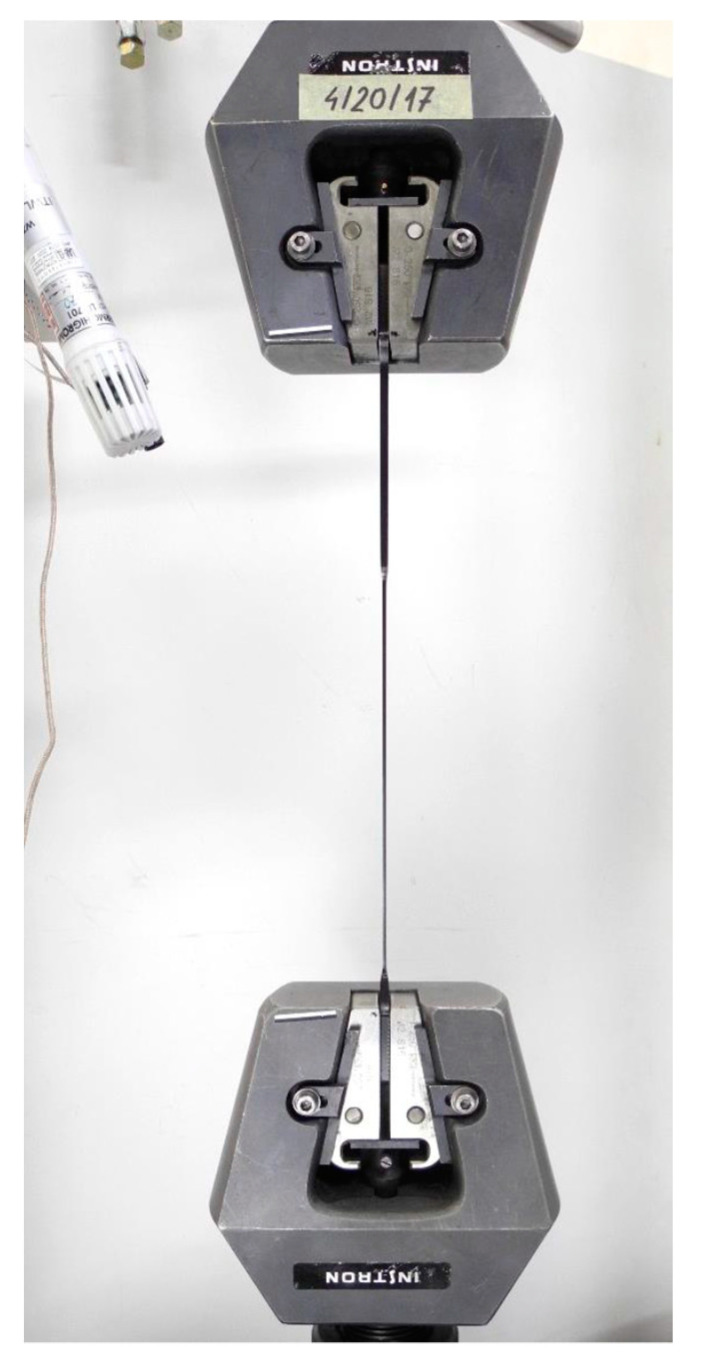
View of the sample fixed in the strength machine after stopping the test.

**Figure 12 materials-14-03557-f012:**
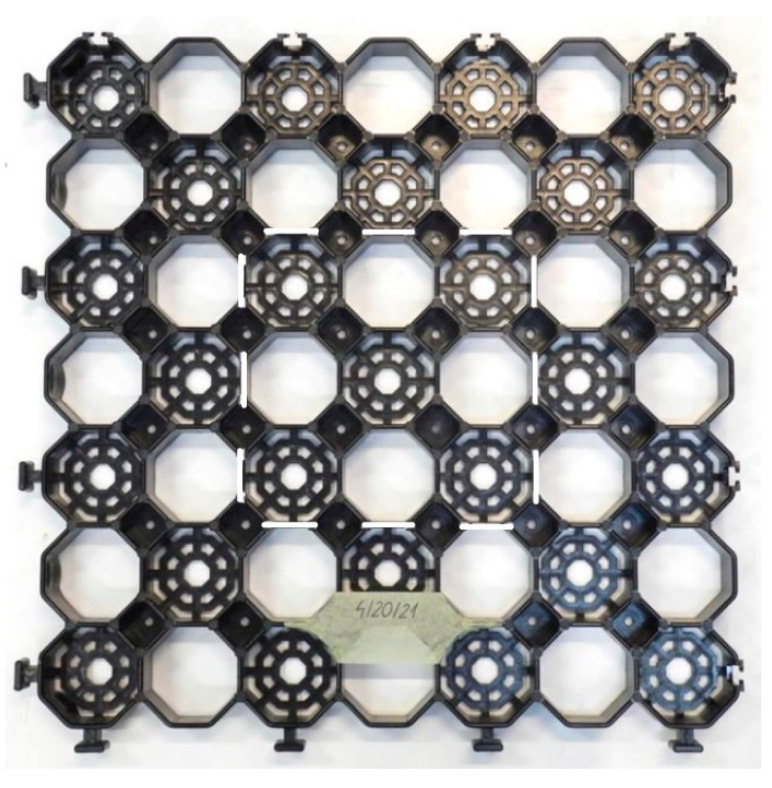
A picture of the geocell before the test; the position of the sample is marked with white lines; compression in the middle area.

**Figure 13 materials-14-03557-f013:**
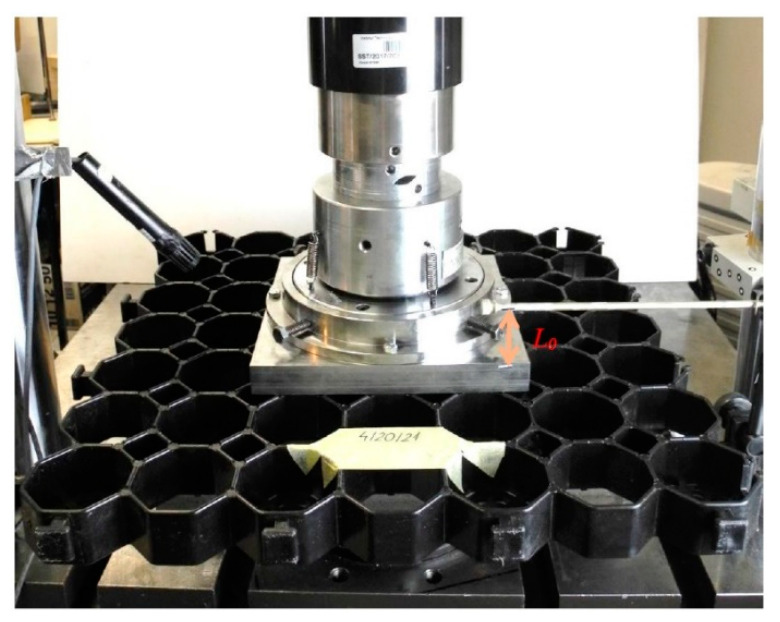
View of the geocell fixed in the strength machine.

**Figure 14 materials-14-03557-f014:**
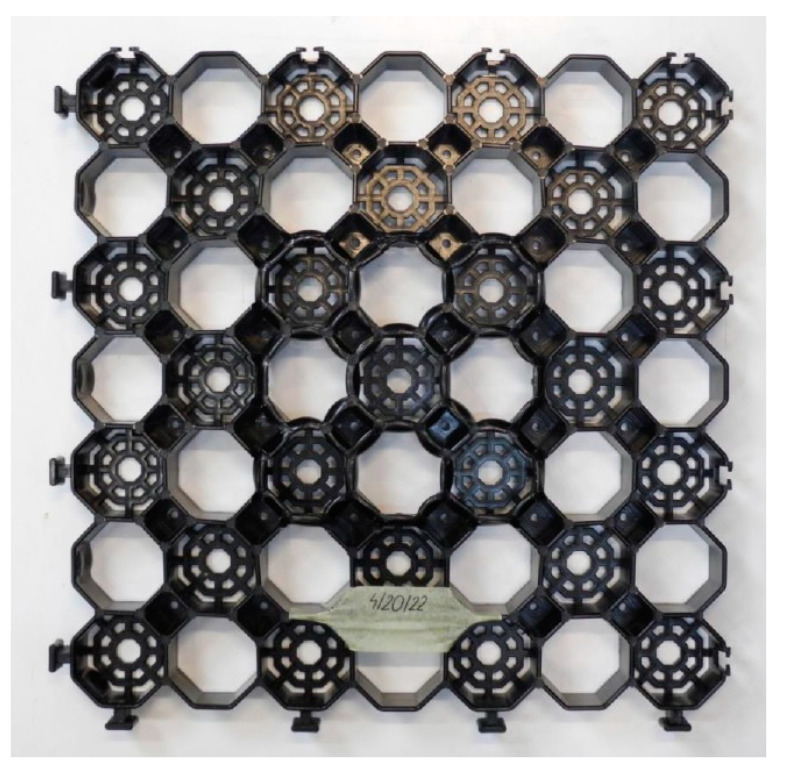
Image of the geocell after the test.

**Figure 15 materials-14-03557-f015:**
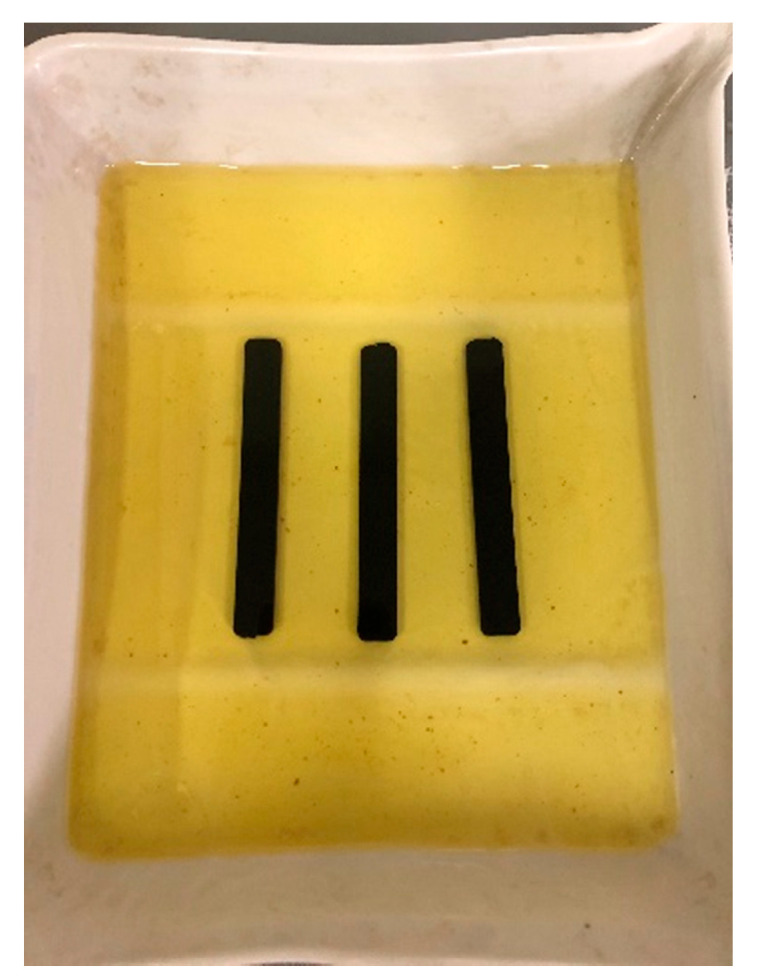
Plastic samples immersed in aviation fuel.

**Figure 16 materials-14-03557-f016:**
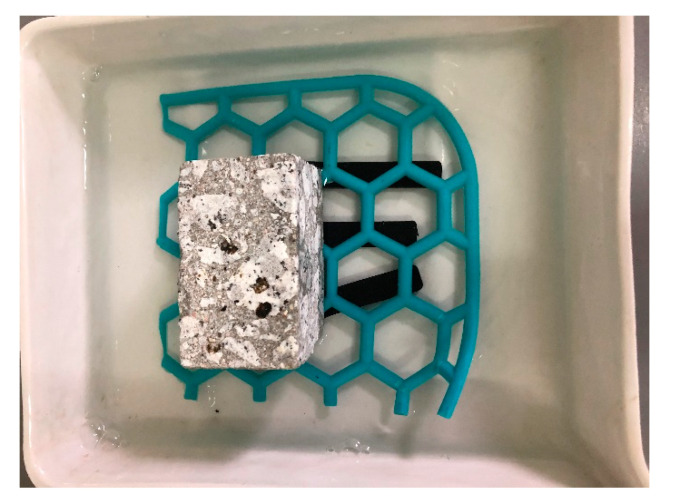
Plastic samples immersed in water.

**Figure 17 materials-14-03557-f017:**
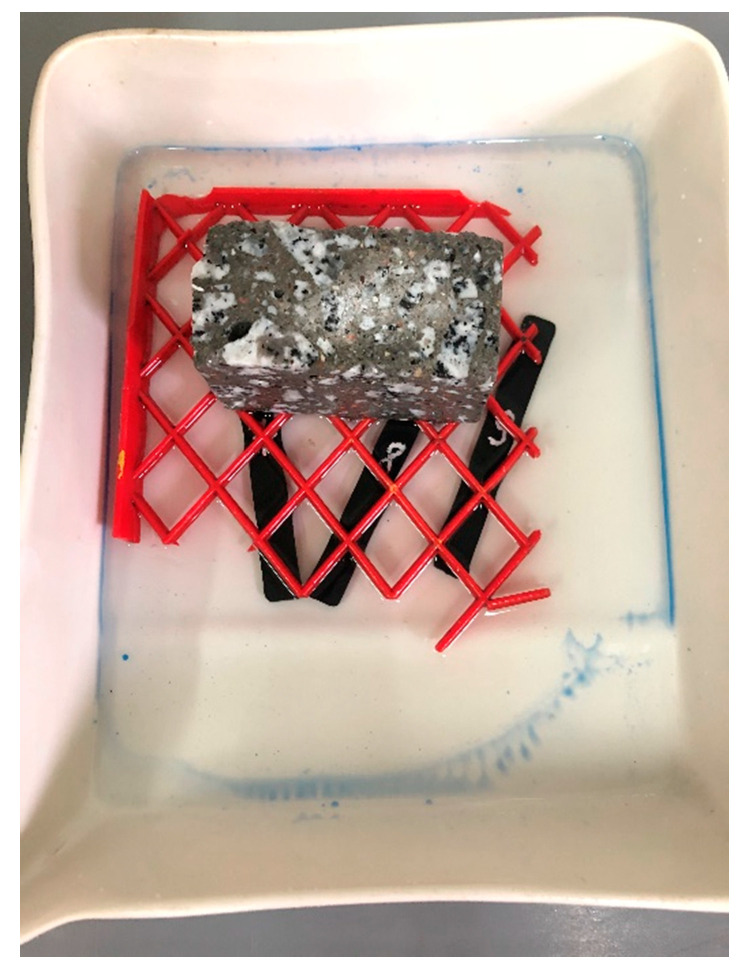
Plastic samples immersed in de-icing agent.

**Figure 18 materials-14-03557-f018:**
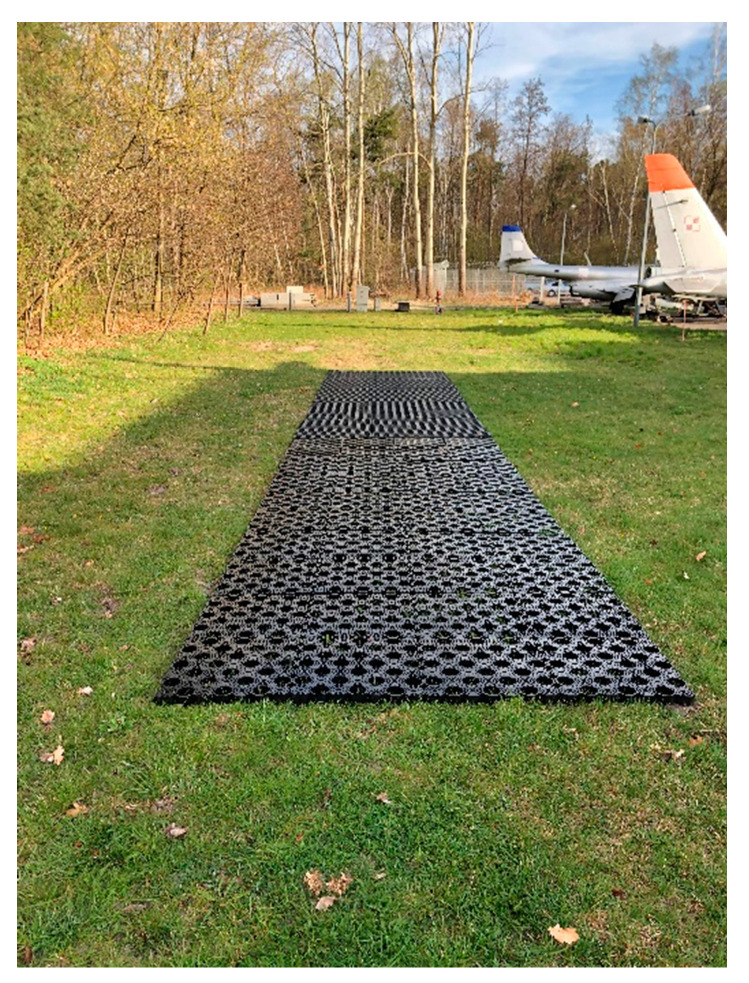
Airfield geocell ready to be pressed into the pavement.

**Figure 19 materials-14-03557-f019:**
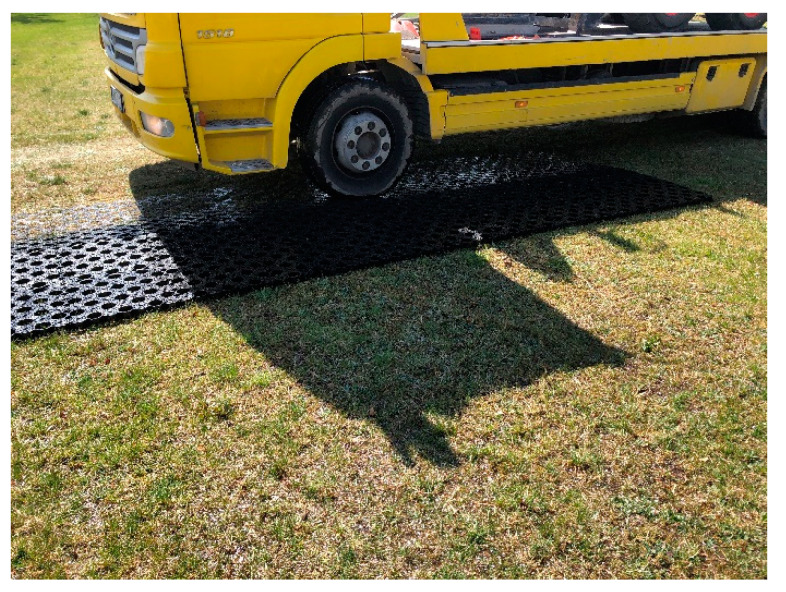
The process of pressing the airfield geocell into the natural airfield pavement.

**Figure 20 materials-14-03557-f020:**
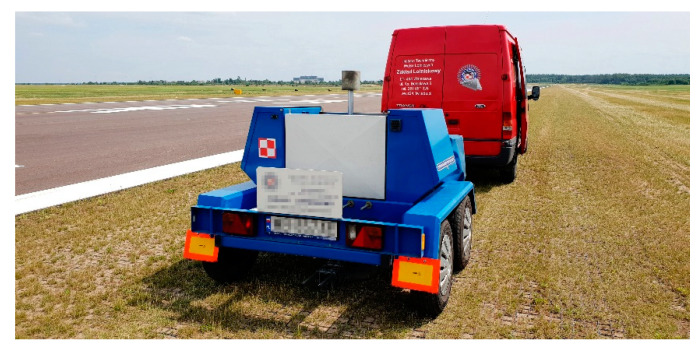
Measurement of load carrying capacity with HWD airport deflectometer.

**Figure 21 materials-14-03557-f021:**
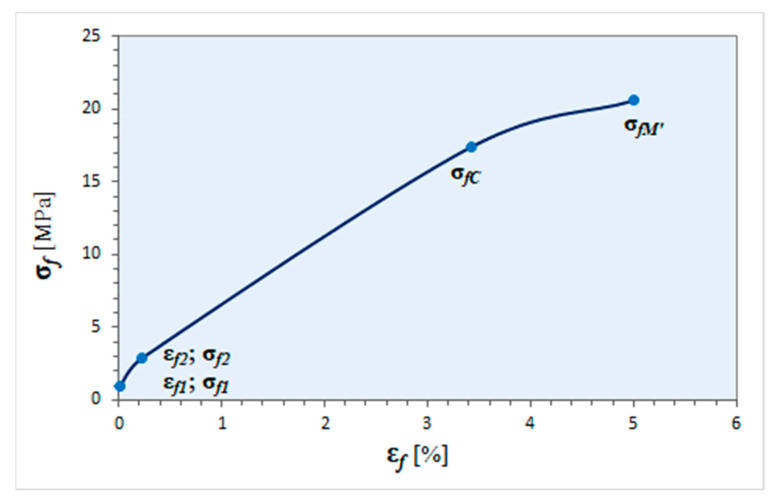
Graph of stress σ_f_ to strain ε_f_ (marked: points for determining the elasticity module at flexural, stress at conventional bend and maximum stress at deformation ε_f_ = 5%).

**Figure 22 materials-14-03557-f022:**
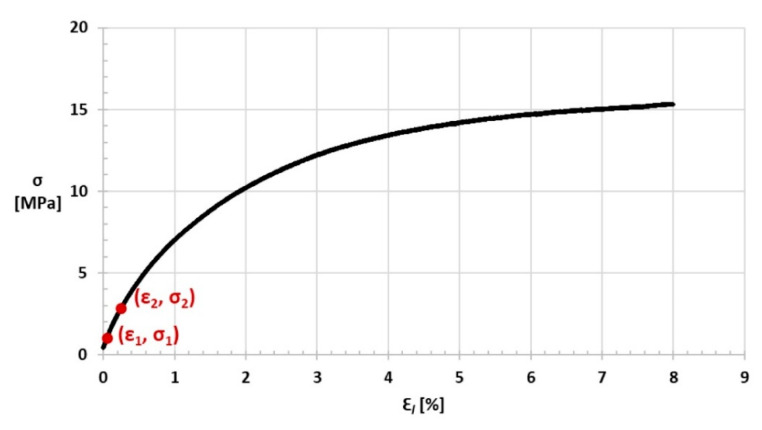
Stress σ relation to ε_l_ strain graph (points to determine the tensile modulus E_t_ are selected).

**Figure 23 materials-14-03557-f023:**
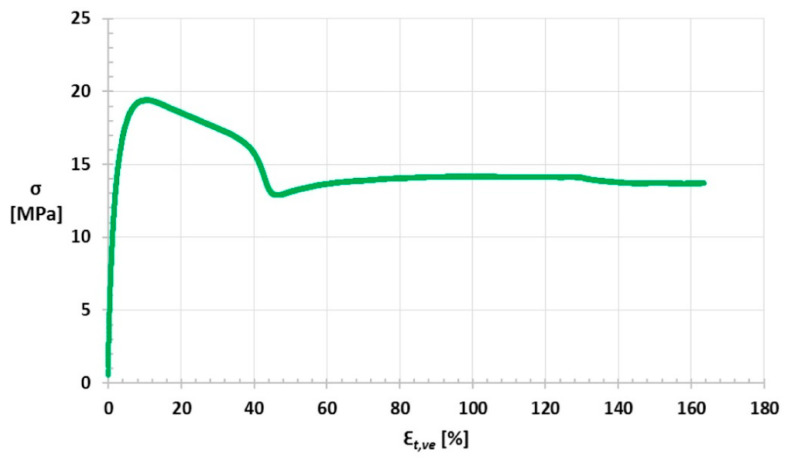
Stress σ relation to strain ε_t,ve_ graph.

**Figure 24 materials-14-03557-f024:**
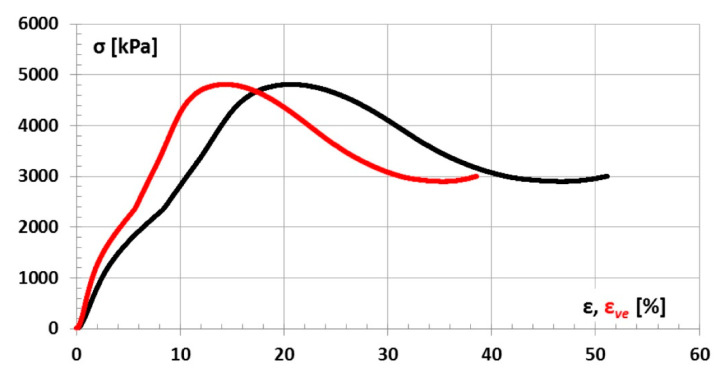
Stress σ relation to strain ε and ε_ve_ graph.

**Figure 25 materials-14-03557-f025:**
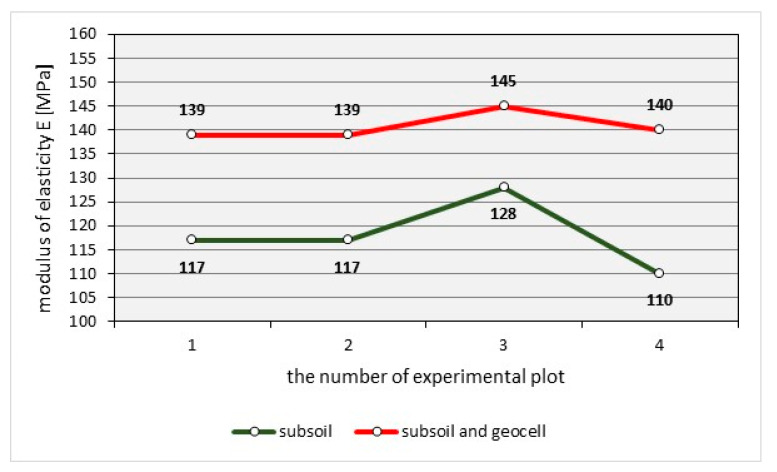
Comparison of the values of the modulus of elasticity obtained on the subsoil and on the structure of the natural pavement with airfield geocell.

**Table 2 materials-14-03557-t002:** Results of flexural stress determination at conventional bend σ_fC_ for a population of 8 samples.

Sample Number	σ_fC_ ^1^ [MPa]	σ_fCśr_ [MPa]	S_(σfC)_ ^2^ [MPa]	m_σfC_ ^3^ [MPa]	S_C_ ^4^ [mm]	ε_fC_ ^5^ [%]
1	17.6	17.5	0.14	17.4 < m_σfC_ < 17.6	5.938	3.44
2	17.6				5.925	3.43
3	17.5				5.952	3.46
4	17.3				5.958	3.46
5	17.4				5.918	3.42
6	17.7				5.928	3.43
7	17.4				5.924	3.42
8	17.6				5.934	3.44

^1^ flexural stress at the conventional deflection S_C_; ^2^ standard deviation of flexural stress at conventional bending S_C_; ^3^ 95% two-sided confidence interval of average values; ^4^ conventional deflection; ^5^ flexural deformation for conventional deflection S_C_.

**Table 3 materials-14-03557-t003:** Results of determination of the elasticity modulus under tension E_t_ for a population of 5 samples.

Sample Number	E_t_ ^1^ [MPa]	E_t,śr_ [MPa]	S_(Et)_ ^2^ [MPa]	m_Et_ ^3^ [MPa]
1	900	913	26.5	889 < m_Et_ < 939
2	890			
3	897			
4	955			
5	923			

^1^ tensile modulus, modulus of elasticity under tension; ^2^ standard deviation of the modulus of elasticity E_t_; ^3^ 95% two-sided confidence interval of average values.

**Table 4 materials-14-03557-t004:** Poisson’s number determination results µ for a population of 5 samples.

Sample Number	μ ^1^ [-]	μ_śr_ [-]	S_(_μ_)_ ^2^ [-]	m_μ_ ^3^ [-]
1	0.49	0.494	0.0	0.48 < m_μ_ < 0.50
2	0.49			
3	0.49			
4	0.50			
5	0.50			

^1^ Poisson’s number; ^2^ standard deviation of Poisson’s number μ; ^3^ 95% two-sided confidence interval of average values.

**Table 5 materials-14-03557-t005:** Results of the determination of the maximum stress σ_m_ for a population of 5 samples.

Sample Number	σ_m_ ^1^ [MPa]	σ_m,śr_ [MPa]	S_(σm)_ ^2^ [MPa]	m_σm_ ^3^ [MPa]
1	19.4	19.8	0.34	19.4 < m_σm_ < 20.2
2	20.0			
3	19.4			
4	20.1			
5	19.9			

^1^ stress at the first local maximum observed during a tensile test; ^2^ standard deviation of strength σ_m_; ^3^ 95% two-sided confidence interval of average values.

**Table 6 materials-14-03557-t006:** Strain results at maximum stress ε_tm,ve_ for a population of 5 samples.

Sample Number	ε_tm,ve_ ^1^ [%]	ε_tm,ve,śr_ [%]	S_(_ε_tm,ve)_ ^2^ [%]	m_εtm,ve_ ^3^ [%]
1	10.2	10.1	0.40	9.7 < m_εtm.ve_ < 10.5
2	9.5			
3	10.5			
4	9.8			
5	10.3			

^1^ nominal strain in the longitudinal direction at maximum stress σ_m_; ^2^ standard deviation of nominal strain in the longitudinal direction at maximum stress ε_tm,ve_; ^3^ 95% two-sided confidence interval of average values.

**Table 7 materials-14-03557-t007:** Stress determination results at the end of test σ_b’_ for a population of 5 samples.

Sample Number	σ_b’_ ^1^ [MPa]	σ_b’,śr_ [MPa]	S_(_σ_b’)_ ^2^ [MPa]	m_σb’_ ^3^ [MPa]	ε_tb,ve’_ ^4^ [%]
1	14.1	14.2	0.29	13.9 < m_σb’_ < 14.5	160
2	14.4				
3	13.7				
4	14.4				
5	14.2				

^1^ stress at break; ^2^ standard deviation of stress at the end of the test σ_b’_; ^3^ 95% two-sided confidence interval of average values; ^4^ nominal strain in the longitudinal direction at σ_b’_ stress; determined if the sample has not broken.

**Table 8 materials-14-03557-t008:** Results of determination of compressive strength σ_mr_.

Sample Number	σ_mr_ [kPa]	σ_mr,śr_ [kPa]	S_(σmr)_ [kPa]
1	4836	4811	27.9
2	4817		
3	4781		
1-1	4110	-	
1-2	3759		
1-3	4082		
1-4	4287		

**Table 9 materials-14-03557-t009:** Results of determination of strain at compression ε_mr_.

Sample Number	ε_mr_ [%]	ε_mr,śr_ [%]	S_(εmr)_ [%]
1	20	20	0.6
2	21		
3	20		
1-1	19	-	
1-2	18		
1-3	18		
1-4	19		

**Table 10 materials-14-03557-t010:** Results of determination of strain at compression ε_mr,ve_.

Sample Number	ε_mr,ve_ [%]	ε_mr,ve,śr_ [%]	S_(εmr,ve)_ [%]
1	14	14	0.0
2	14		
3	14		
1-1	16	-	
1-2	16		
1-3	15		
1-4	16		

**Table 11 materials-14-03557-t011:** Summary of plastic resistance test results after 14 days of soaking in consumables.

Medium	Sample Number	Weight before Soaking m [g]	Weight after Soaking m_1_ [g]	Weight Change [%]	Absorption Average Value [%]
Water	1	2.935	2.938	0.003	0.14 s = 0.035 v = 25.3
	2	2.913	2.918	0.005	
	3	2.919	2.923	0.004	
De-icing agent	4	2.94	2.944	0.004	0.11 s = 0.039 v = 34.6
	5	2.933	2.935	0.002	
	6	2.933	2.937	0.004	
Aviation fuel	7	2.935	3.029	0.094	3.17 s = 0.038 v = 1.2
	8	2.939	3.031	0.092	
	9	2.92	3.013	0.093	

**Table 12 materials-14-03557-t012:** Typical properties of the plastic from which the airfield geocell is made of.

Typical Properties	Nominal Value	Unit	Test Method
Flow rate indicator (MFR) 190 °C/2.16 kg 190 °C/5.0 kg	4.0 11.0	g/10 min g/10 min	ISO 1133-1 [40]
Density	0.955	g/cm^3^	ISO 1183-1 [41]
Tensile flexibility modulus	1200	MPa	ISO 527-1, -2 [31,42]
Stress at the yield strength limit	27	MPa	ISO 527-1, -2
Elongation at the yield strength	8	%	ISO 527-1, -2
FNCT (3.5 MPa 2% Arkopal N100 80 °C)	4.5	h	ISO 16770 [43]
Toughness with notch according to Charpy’ego 23 °C, Type 1, karb A −30 °C, Type 1, karb A	4.0 4.5	kJ/m^2^ kJ/m^2^	ISO 179 [44]
Shore Hardness (Shore D)	60	-	ISO 868 [45]
Ball hardness (H 132/30)	52	MPa	ISO 2039-1 [46]
Vicata softening temperature (B/50 N)	73	°C	ISO 306 [47]

## Data Availability

The data presented in this study are available on request from the corresponding author.

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
