# Peer review of "Analysis and Assessment of the Usefulness of Recycled Plastic Materials for the Production of Airfield Geocell"

_materials, 2021, doi:10.3390/ma14133557_

Round 1

Reviewer 1 Report

Dear authors,

the paper is of scientific and original nature, related to Analysis and assessment of the usefulness of recycled plastic materials for the production of airfield geocell.

For a better clarification, please edit your paper as follows:

Enlarge the Introduction with current results reported in the world and Europe, - References to expand the results of European authors registered in SCOPUS / WoS such as: Analysis of diagnostic methods and energy of production systems drives. Figures no. 5, 15  - 17 should be contrasting and readable and insert according to the instructions in the template. Figures 5 and 6 – the description is not clear. The diagrams (Fig. 21 - 25) are unreadable. Several indexes in the whole article are unreadable, please check. Reference number 9 is missing in the text, please complete them. Authors should check the whole text, figures, references and improve them, according to journal’s guidelines, eg. lack of DOI in the references.

Conclusions and future work should be extended to contain practical applications based on research described in this paper, edit the paper according to the template. Please revise the manuscript with English grammar. There are many places that the manuscript needs to be improved with respect to English writing. Please, edit the paper according to previous comments and after minor changes I recommend the paper to be published.

Author Response

  1. In Europe and in the world, airfield geocell is used to reinforced natural airfield pavements, however, there are no documented results in the literature.
  2. The proposed article is not related to the article under evaluation.
  3. Figures no. 5 and 15-17 corrected.
  4. Figures 5 and 6 – the description corrected.
  5. Figures 21-25 – resolution has been increased.
  6. Reference number 9 is in a chapter 1.2.
  7. We added DOI in the references.

Reviewer 2 Report

Dear Authors,

This paper shows a very interesting topic that is essential to recycle plastic and use in the airfield [pavements. The structure is well addressed and the paper, in general, showing a good agreement of the journal instructions. Thanks for your efforts and work done.

I have the following comments on the paper before considering it for publication. I would like that, if possible, authors can clarify and address the comments below; 

Comments:

1- It would be good if the authors show the development of this approach and compared it to previous geocells for example geoglass, geocarbon, and geotextiles … etc? Why your approach can be significant to achieve better results or an accurate result?

2- Figures needs to be interpreted on figures and captions, all figures are not well described nor features are interpreted.

3- I will suggest to do separate two sections:

    a- Purpose of the methodology ------- add to introduction part.

    b- Advantages and limitation of these geocells ------- could be includes as a section

        with general conclusions.

4- Abstract is a bit confusing, as the reader will not be fully understanding why this approach is contributed? Why not others and what can be replaced from SoA.  

5- Add some explanations about assessment and monitoring of such geocells for example after 10 years of installing, in SoA or conclusion as limitations … etc

Please consider reading these articles and some more:

NDT assessment of rigid pavement damages with Ground Penetrating Radar: Laboratory and field tests. International Journal of Pavement Engineering, Taylor and Francis, 2020. https://doi.org/10.1080/10298436.2020.1778692

6- Please, re-write the conclusion and reflect on your results. Besides, explain more about the possible advantages and disadvantages of this approach (refer to comment #2), considering the limitations of your results is also significant to further studies?

Please consider re-review according to the above-mentioned comments, I suggest a major revision of the paper.

Looking forward to receiving the revised version.  

Author Response

  1. We have added an explanation in point 1.2.
  2. Where necessary, figures are described.
  3. We believe the current layout is appropriate.
  4. The summary has been supplemented.
  5. The explanation is described in chapter 5. The article was referenced in the text.

Reviewer 3 Report

1) Please, increase the resolution of figures 5-6.
2) Insert the sections "Discussion" and "Conclusion" after Results.
3) It would be nice if you could add and a section Further developments.
4) Insert more references especially from the last three years.

Author Response

  1. We increased the resolution of figures 5-6.
  2. We inserted the section „Discussion” and the section „Conclusion”.
  3. We described added Further developments in chapter.
  4. There are enough references in the article so we leave it as it is.

Round 2

Reviewer 2 Report

Thank you for addressing all comments.

Reviewer 3 Report

The work was much improved and arranged, reaching a proper final form. The figures have all been arranged and now look very good. All aspects and missing chapters have been added. In its current revised form, the paper can be accepted directly for publication.